# Temporal Dynamics of the Gut Bacteriome and Mycobiome in the Weanling Pig

**DOI:** 10.3390/microorganisms8060868

**Published:** 2020-06-09

**Authors:** Ann M. Arfken, Juli Foster Frey, Katie Lynn Summers

**Affiliations:** Animal Biosciences and Biotechnology Laboratory, U.S. Department of Agriculture, Agricultural Research Service, Beltsville, MD 20705, USA; ann.arfken@usda.gov (A.M.A.); juli.frey@usda.gov (J.F.F.)

**Keywords:** piglet, weaning, microbiome, mycobiome, bacteriome, porcine

## Abstract

Weaning is a period of environmental changes and stress that results in significant alterations to the piglet gut microbiome and is associated with a predisposition to disease, making potential interventions of interest to the swine industry. In other animals, interactions between the bacteriome and mycobiome can result in altered nutrient absorption and susceptibility to disease, but these interactions remain poorly understood in pigs. Recently, we assessed the colonization dynamics of fungi and bacteria in the gastrointestinal tract of piglets at a single time point post-weaning (day 35) and inferred interactions were found between fungal and bacterial members of the porcine gut ecosystem. In this study, we performed a longitudinal assessment of the fecal bacteriome and mycobiome of piglets from birth through the weaning transition. Piglet feces in this study showed a dramatic shift over time in the bacterial and fungal communities, as well as an increase in network connectivity between the two kingdoms. The piglet fecal bacteriome showed a relatively stable and predictable pattern of development from *Bacteroidaceae* to *Prevotellaceae*, as seen in other studies, while the mycobiome demonstrated a loss in diversity over time with a post-weaning population dominated by *Saccharomycetaceae*. The mycobiome demonstrated a more transient community that is likely driven by factors such as diet or environmental exposure rather than an organized pattern of colonization and succession evidenced by fecal sample taxonomic clustering with nursey feed samples post-weaning. Due to the potential tractability of the community, the mycobiome may be a viable candidate for potential microbial interventions that will alter piglet health and growth during the weaning transition.

## 1. Introduction

The gastrointestinal (GI) tract is home to the gut microbiome, trillions of colonizing and transient microbes. These microbes support critical health functions including digestion, immune development, metabolism, and resistance to pathogens. While many studies have focused on the bacterial component of the microbiome, the bacteriome, recent studies have demonstrated the ability of fungal microbiome members, the mycobiome, to alter gut microbial community structure and cause disease [1,2,3,4,5]. Commensal fungi can alter the severity of disease as well as modify host immunological responses during normal health [6,7,8,9]. In pigs, the diversity trends of the mycobiome and the complex interactions between the indigenous bacteria and fungi in the gut remain largely unknown, as most studies have investigated pathogenic fungi or mycotoxins, fungal secondary metabolites.

The weaning transition involves drastic changes in diet and environment, resulting in stress and broad alterations in the microbiome. These changes can result in poor growth performance and a predisposition to infections, such as post-weaning diarrhea, making potential health interventions of interest to industry and farmers [10,11,12]. Previous work from our laboratory investigated the bacterial and fungal members of the piglet mucosal-associated gastrointestinal tract microbiome two weeks post-weaning (day 35) to characterize the healthy microbiome. Bacterial and fungal populations were distinct between GI organs (stomach, duodenum, jejunum, ileum, cecum, and colon) and our data suggested that some fungal species had positive and negative correlations with certain bacterial groups [13]. *Kazachstania slooffiae* is a commonly found fungus in pigs and is proposed to be a commensal [13,14,15,16] while other fungi, such as *Aspergillus*, have negative inferred interactions with beneficial bacteria like *Prevotella* [13]. However, this study only provided information on the mycobiome at a single time point (day 35) and the development of these important microbial populations remains poorly understood. The aim of this study was to analyze the temporal development of both the fecal microbiome and mycobiome from birth through two weeks post-weaning in healthy piglets to help gain insight into potential dietary interventions during this critical period.

## 2. Materials and Methods

### 2.1. Animals

In total, 23 Large White x Landrace piglets from 3 litters (L.1–3) were assessed from birth through day 35 of age and were weaned at day 21. Piglets and lactating sows were housed in a single farrowing barn located at the USDA-ARS facility in Beltsville, MD. Adult pigs are housed in outdoor housing prior to farrow, when gilts and sows are brought into farrowing facilities. Piglets were not provided milk replacer/supplement or creep feed at any point throughout the experiment. The diet was formulated to meet the National Research Council estimate of nutrient requirements (Appendix A). Following weaning at day 21, piglets were moved to pens located in a single nursery room connected to the farrowing barn and separated by litter. From days 21 to 28, piglets received Nursery Diet 1 followed by Nursery Diet 2 from days 29 to 35. Piglets were given free access to feed and water, evaluated daily for health and weights were measured for each sample time point (Appendix A); all piglets used in this study were observed to be healthy. No antibiotics, antifungals, or supplementary additives were administered to the piglets at any time during the experiment. Care and treatment of all pigs were approved by the USDA-ARS Institutional Animal Care and Use Committee of the Beltsville Agricultural Research Center (AUP#18-022).

### 2.2. Sample Collection

Fresh fecal samples were collected into sterile cryovial tubes from the rectum of individual piglets (*n* = 23) for the following 8 time intervals: days 1 (post-natal), 3, 7, 14, 21, 24, 28 and 35. Additional fresh fecal samples were collected from indoor lactating sows (*n* = 2), adult female pigs (*n* = 5 pigs/pen) housed in outdoor pens at the Beltsville facility (labeled A–F), and nursery feed samples (*n* = 3). Samples were initially flash frozen in liquid nitrogen and stored at −80 °C until further processing. A total of 234 samples were collected.

### 2.3. DNA Extraction and Sequencing

DNA was isolated from 0.25 g feces using the MagAttract Power Microbiome Kit (Qiagen, Hilden, Germany) by the Microbial Systems Molecular Biology Laboratory at the University of Michigan. Cells were lysed to isolate DNA using mechanical bead beating for 20 total minutes with 20 frequency/second and extracted using magnetic bead technology according to the Qiagen protocol. The V4 region of the 16S rRNA-encoding gene was amplified from extracted DNA using the barcoded dual-index primers developed previously [17]. The ITS region was sequenced utilizing primers ITS3 (5′ GCATCGATGAAGAACGCAGC 3′) and ITS4 (5′ TCCTCCGCTTATTGATATGC 3′) with the Illumina adaptor sequence added to the 5′ end. Both the 16S and ITS regions were sequenced with the Illumina MiSeq Sequencing platform, generating 250 and 300 bp paired-end reads, respectively. All sequences are publicly available in the NCBI’s Sequence Read Archive (SRA) under accession IDs PRJNA613280 and PRJNA610764.

### 2.4. Bacteriome and Mycobiome Sequence Processing

#### 2.4.1. Bacteria (16S)

Quality filtering, pairing, denoising, amplicon sequence variant (ASV) determination, and chimera removal were conducted with the DADA2 plugin [18] in QIIME2 v. 2019.7 [19]. For quality trimming, paired-end sequences were truncated to 240 and 160 bp for forward and reverse reads, respectively. Taxonomic classification of the ASVs was performed using the pretrained 16S 515F/806R from the Silva 132 database [20]. ASVs identified as Archaea, chloroplast, mitochondria, or unassigned were removed from further analysis.

#### 2.4.2. Fungi (ITS)

Forward and reverse primers were removed from paired-end reads with cutadapt v 1.18 [21]. Due to the biologically relevant length variation found in fungal ITS sequences, Trimmomatic v 0.39 [22] and the sliding window option were used to trim individual sequences where the average quality score was <15 across 4 base pairs. Following primer removal and trimming, QIIME2 plugin DADA2 was used to identify ASVs. Taxonomic classification was trained and conducted on fungal sequences using the UNITE [23] developer’s full-length, dynamic ITS reference sequences in QIIME2. Fungal ASVs without a phylum or higher classification or those identified as unassigned were removed. Additional classification using BLAST (https://blast.ncbi.nlm.nih.gov) (National Center for Biotechnology Information, Bethesda, MD, USA) was performed on unassigned sequences to confirm non-fungal origin.

Separate rarefaction curves for bacterial and fungal samples were produced using the vegan package [24] in R v 3.5.1 (R Core Team, 2018 https://www.R-project.org) and visualized in GraphPad Prism v 7 (La Jolla, CA, USA) to determine minimum sequencing depth (Appendix A). A cutoff of 5000 sequences was determined for bacterial and fungal samples. Samples <5000 sequences were removed (bacteria, *n* = 23; fungus, *n* = 8). A total of 5,369,707 sequences from 211 bacterial and 17,339,872 sequences from 226 fungal samples were selected for downstream analysis (Appendix A).

#### 2.4.3. Characterization of the Bacteriome and Mycobiome

Calculations of alpha diversity including Shannon diversity, evenness and observed ASVs among piglet fecal samples were performed on rarefied bacterial (*n* = 5433) and fungal (*n* = 5144) samples using the phyloseq package [25]. Satisfaction of normality was tested using the Shapiro–Wilk test, and data was transformed using box cox root transformations when necessary. A mixed-effects linear model with piglet as the random variable was used to determine significant trends in diversity over time. Non-metric multidimensional scaling (NMDS) of piglet, adult, and feed communities were conducted using the vegan package on log-transformed bacterial and fungal sequence counts using Bray–Curtis dissimilarity distances. To reduce potential ASV artifacts, ASVs present in <1.0% of samples were removed prior to analysis. NMDS of pre- and post-wean piglet fecal communities were calculated as described above. The envfit function in the R-package vegan was used to fit taxonomic vectors to the pre- and post-wean ordination plots [24]. NMDS plots were visualized using the ggplot2 package [26]. Permutational analysis of variance (PERMANOVA) was used to determine the main effect of growth time points on the piglet fecal microbial communities using the adonis function in vegan. The strata option (strata = piglet) was used to account for repeated sampling of individual piglets. Pairwise comparisons of mean Bray–Curtis distances to group centroids among piglet fecal samples were calculated using the permutational analysis of multivariate dispersion (PERMDISP) function in vegan and plotted in R. Relative abundances of taxa are presented as the mean % value by litter for each fecal sampling time point. Differentially abundant genera between pre- and post-wean piglet fecal communities were determined using the linear regression log (llm2) method implemented in the R-package DAtest [27], with piglet as the paired variable and litter as a covariate. Figures for taxonomy bar plots and differential abundance plots were created with GraphPad Prism 8. Co-occurrence network analysis to test for associations between bacterial and fungal genera were conducted using the Meinhausen–Buhlmann neighborhood selection method in the Sparse and Compositionally Robust Inference of Microbial Ecological Networks (SPIEC-EASI) [28] R-package. Genera present in <20% of the samples for each time point of growth were removed prior to analysis to reduce noise and complexity. Networks were plotted using the R-package igraph [29]. Unless otherwise indicated, errors are given as ± SE and *p*-values were corrected for multiple testing using FDR, and significance was determined as *p* < 0.05.

## 3. Results

### 3.1. Temporal Diversity Trends in the Bacteriome and Mycobiome

The piglet fecal bacteriome and mycobiome showed different temporal trends in alpha diversity over the eight time points from day 1 through day 35 post-natal (Figure 1 and Appendix A). In the bacteriome, microbial diversity (Shannon index; β_1_ = 0.11, SE = 0.18, *p* < 0.001), richness (observed ASVs; β_1_ = 6.44, SE = 0.30, *p* < 0.001) and evenness (β_1_ = 0.002, SE = 0.0001, *p* < 0.001) all demonstrated overall significant positive increases from day 1 to day 35, with the greatest increases between day 1 to day 7 and day 21 to day 24 (Figure 1A,C and Appendix A). Alpha diversity plateaued or showed a slight decrease following day 24. In contrast, the mycobiome demonstrated a decrease in diversity (Shannon index; β_1_ = −0.01, SE = 0.003, *p* < 0.001) and evenness (β_1_ = −0.004, SE = 0.0006, *p* < 0.001) from day 1 to day 35, with most of the decrease occurring post-wean (Figure 1B,D). Richness (observed ASVs; β_1_ = 0.01, SE = 0.002, *p* < 0.001) in the mycobiome showed a slight increase from day 1 to day 35, with the greatest increase occurring post-wean from day 21 to day 28 (Appendix A).

Non-metric multidimensional scaling (NMDS) plots based on Bray–Curtis distance of the piglet fecal microbiomes revealed significant temporal shifts from day 1 to day 35 in both the bacteriome (PERMANOVA; F = 22.25, R^2^ = 0.49, *p* < 0.001) and mycobiome (PERMANOVA; F = 8.17, R^2^ = 0.25, *p* < 0.001) (Figure 2, Appendix A). In the bacteriome, each of the eight time points from day 1 to 35 showed distinct clusters and centroids, with the largest separation between clusters occurring between pre- and post-wean samples (Figure 2A). The outdoor adult and lactating sow fecal bacteriome was closely clustered with the post-wean samples, but remained distinct from the piglet fecal bacteriome, while the feed sample did not cluster closely with any of the time points. Dispersion of samples showed a decreasing trend from day 1 to day 35, with day 35 having significantly less dispersion (PERMDISP; *p* < 0.001) than the rest of the time points (Figure 3A, Appendix A). A similar, but slightly less-distinct, temporal clustering pattern was displayed in the piglet fecal mycobiome (Figure 2B). The greatest separation of clusters occurred between pre- and post-wean samples; however, there was more overlap of samples among time points, particularly among pre-wean piglet samples. Unlike the bacteriome, the feed samples clustered near outdoor adult and piglet day 35 fungal samples. Dispersion among the fecal mycobiome samples significantly increased from day 1 to day 7 (PERMDISP; *p* < 0.001), then decreased to similar dispersion levels found at day 1 (Figure 3B, Appendix A).

### 3.2. Evaluation of the Bacteriome and Mycobiome Composition

The dynamic shifts in bacteriome and mycobiome taxonomic composition over the eight time points (day 1 to day 35) were examined at the phylum, family and genus classification level. At the phylum level, the bacteriome transitioned from a Proteobacteria- to a Firmicutes- and Bacteroidetes-driven community (Appendix A). At the family level, a dramatic shift in taxa was demonstrated from the pre-wean (day 1–21) to post-wean (day 24–35) time period, with post-wean family taxa similar to those found in the sows and adult pigs (Figure 4A). The dominant (>60% of sequences) families, *Staphylococcaceae* and *Micrococcaceae*, found in nursery feed were found in <0.10% of piglet fecal sequences. Family taxon vectors fitted to an NMDS ordination plot displaying pre- and post-wean bacteriomes revealed that dominant (>50% of sequences) families *Enterobacteriaceae*, *Enterococcaceae*, *Fusobacteriaceae*, and *Bacteroidaceae* were significantly correlated with pre-wean piglet feces, while *Veillonellaceae*, *Prevotellaceae*, *Lactobacillaceae*, *Ruminococcaceae* and *Lachnospiraceae* were significantly correlated with post-wean piglet feces (*p* < 0.05, Appendix A). Differentially abundant genera between pre- and post-wean piglet fecal bacteria included highly abundant *Bacteroides* (*p* < 0.001), *Clostridium* (*sensu stricto* 1 and 2, *p* < 0.001), and *Escherichia* (*p* < 0.001) in pre-wean piglet feces and *Blautia* (*p* < 0.001), *Prevotella* (1, 2 and 9, *p* < 0.001) and *Ruminococcus* 1 (*p* < 0.001) in post-wean piglet feces (Figure 5A–F, Appendix A).

In the mycobiome, there was an overall shift from Mucoromycota and Basidiomycota pre-wean (day 1 to day 21) to Ascomycota post-wean (day 24–day 35) in piglet feces (Appendix A). At the family level, dominant (>50% of sequences) families *Trichosporonaceae*, *Symbiotraphinaceae*, *Mucoraceae*, and *Cladosporaceae* significantly correlated (*p* < 0.05) with pre-wean piglet feces (Appendix A) and *Saccharomycetaceae*, *Debaryomycetaceae* and *Wallemiaceae* significantly correlated (*p* < 0.05) with post-wean piglet feces (Appendix A). Families *Saccharomycetaceae*, *Dipodascaceae*, *Aspergillaceae*, *Debaryomycetaceae*, and *Wallemiaceae*, making up >43% of the sequences found in nursery feed samples, were also present in >79% of sequences in post-wean piglet feces and <18% of pre-wean piglet feces (Figure 4B). Sows and adult pigs showed slightly different taxonomic profiles than post-wean piglets, with an overall greater mean relative abundance of *Dipodascaceae* (11.7 ± 3.5% vs. 4.6 ± 0.9%) and *Neocallimastigaceae* (10.3 ± 1.4% vs. 3.1 ± 1.4%) and a lower mean relative abundance of *Wallemia* (2.6 ± 0.4% vs. 17.0 ± 2.0%). Genera *Mucor* (*p* < 0.001), *Cladosporium* (*p* < 0.05), and *Trichosporon* (*p* < 0.001) were significantly more abundant in pre-wean piglet feces, while *Wallemia* (*p* < 0.001), *Kazachstania* (*p* < 0.001), and *Hyphopichia* (*p* < 0.001) were significantly more abundant in post-wean piglet feces (Figure 6A–F, Appendix A).

### 3.3. Interactions between the Bacteriome and Mycobiome

A co-occurrence network was constructed to determine the interactions between bacterial and fungal genera in piglet fecal microbiomes at days 1, 21, and 35 (Figure 7A–C). At day 1, only one negative association was found in piglet feces between bacteria *Clostridium sensu stricto* 1 and *Lactobacillus* (Figure 7A). No fungal-bacterial interactions were identified. At day 21, 93 different interactions were determined between the microbiota (Figure 7B). Of the 11 identified fungi present in day 21 feces, four fungal genera showed interactions with bacteria, including a negative association between *Aspergillus* and *Ruminococcaceae* UCG-004 and between *Cladosporium* and *Alistipes*. At day 35, 142 interactions were found in the pig fecal microbiome (Figure 7C). Of the fungal genera, *Hyphopichia* and *Aspergillus* had the highest number of interactions with bacteria (*n* = 4 each), *Hyphopichia* showed positive associations with *Eubacterium rumaniantium* group, *Ruminococcaceae* UCG-003, and *Succiniclasticum* and a negative association with *Escherichia-Shigella*. *Aspergillus* showed a positive association with *Anaerovibrio* and negative associations with *Lachnospiraceae* UCG-010, *Prevotellaceae* UCG-003 and *Subdoligranulum*. Positive fungi-bacteria associations *Penicillium-Lactobacillus*, *Talaromyces*-*Prevotellaceae* UCG-003, *Trichosporon*-*Pygmaiobacter*, and negative fungi-bacteria association *Issatchenkia*-*Lachnospiraceae* UCG-001 were also found.

## 4. Discussion

The colonization, development, and interactions of the piglet gut microbiota from birth through weaning are critical to piglet heath and growth performance. In both bacterial and fungal populations, there is a dynamic shift in community structure and composition during this time, with adult-like microbiomes developing by day 35. Our longitudinal examination of the piglet fecal bacteriome and mycobiome from day 1 (post-natal) through day 35 revealed different patterns in temporal diversity among bacteria and fungus as well as potential interactions in the fecal community.

The temporal alpha diversity patterns found in the piglet fecal bacteriome (Figure 1A) were similar to those found in previous fecal studies [30,31,32,33], with increasing Shannon diversity from birth through the weaning. This same pattern was also evident in evenness and richness over the same time period (Figure 1C and Appendix A). After day 24, diversity in the bacteriome plateaued, suggesting that the progressive early colonization of bacteria coupled with a sudden transition from sow milk to nursery feed resulted in the concurrent, rapid expansion of different bacterial populations, able to exploit novel niches within the piglet GI tract. However, by day 35, these populations appear to have reached stability or equilibrium in the piglet.

In contrast, the fecal mycobiota showed a reduction in Shannon diversity and evenness from birth through weaning (Figure 2B,D). Interestingly, richness remained relatively steady with a slight increase post-weaning (Appendix A). This indicates that Shannon diversity decreased, as a result of reduced evenness, within the mycobiome. Data from our study showed dramatic increases in the presence of two yeast genera, *Kazachstania* and *Hyphopichia*, post-wean (Figure 6), which is likely driving down community evenness by outcompeting other fungal species. In general, high microbial diversity is deemed beneficial to a community or host by providing functional redundancy and resiliency [34,35]. However, in the piglet gut environment, the establishment of dominant yeast species and reduction in community evenness and diversity in healthy, post-weaned piglets may be an exception. In humans, low fungal diversity has been consistently identified in healthy adult feces [36,37], while increased fungal diversity has been linked to gastrointestinal disease [3,38]. We hypothesize that the same may be true in piglets, where the presence of low fungal diversity may indicate health and high fungal diversity may suggest the presence of disease or dysbiosis.

The effect of time on the developing microbiota was clearly evident in both bacterial and fungal populations in the piglet feces, with time being a more important factor in the developing bacteriome than in the mycobiome (Appendix A). In the bacteriome, the effect of time was especially evident in the transition from day 21 to day 24, when piglets were removed from the sow and introduced to nursery feed; this transition was primarily driven by compositional shifts in the bacterial community from milk oligosaccharide-utilizing *Bacteroidaceae* to plant polysaccharide-utilizing *Prevotellaceae*, which has been well documented in other piglet weaning studies [10,31,39,40]. Another important transition in the bacteriome was the immediate reduction in *Enterobacteriaceae* (genus *Escherichia*/*Shigella*) following birth that occurred between day 1 and day 3 (Figure 5). This trend has also been consistently detected in other piglet studies [32,39,41], and is likely part of the similar dynamic transition from aerobes/facultative anaerobes to obligate anaerobes in developing human infant guts [42].

In addition to trends in composition, the bacteriome demonstrated reduced dispersion among communities over time, which was not found in the mycobiome (Figure 3). This indicates that the piglet fecal bacteriome likely follows a defined pattern of colonization and succession in healthy developing piglets, whereas in the mycobiome a large portion of the community may be transient and driven by other factors, such as environmental exposure, diet, or host immunity, which may vary by piglet. Other studies have purported similar hypotheses regarding the transient nature of the mycobiome based on compositional variability among sample subjects and association of fungal genera with food or environmental sources [36,43,44]. In the pre-wean piglet mycobiome, environmental influence is suggested by the dominance of naturally ubiquitous fungal families *Trichosporonaceae*, *Mucoraceae*, and *Cladosporiacea*, all of which are found in a variety of environments including organic debris, soils, and indoor/outdoor air [45,46,47]. Furthermore, post-wean piglet feces mycobiomes clustered near nursery feed samples (Figure 2) and showed a sharp increase in abundance of fungal taxa *Wallemiaceae* and *Debaryomycetaceae*, highly abundant in nursery feed samples, directly following weaning. The role of these transient species on host development and resident bacteria remain to be elucidated. The piglet fecal bacteriomes, in contrast, did not cluster near feed samples nor resemble any of the abundant bacterial taxa found in the nursery feed.

Despite receiving the same feed and housing, there were marked compositional differences between litter mycobiomes by day 35, indicating potential involvement of host genetic or immunity factors in shaping the fungal community [48,49]. One of the most striking differences included a high level of variation in the relative abundances of fungal families *Saccharomycetaceae* (*K. slooffiae*) and *Debaryomcetaceae* (*H. burtonii*). *K. slooffiae*, a persistent fungus in post-weaned piglets [30,44], is hypothesized to be a potential protein source to healthy pigs [14] and may behave similarly to commensal *Candida* species in humans [13]. In a study on human diets, *Candida* was positively correlated with high carbohydrate consumption [50] and has been shown to degrade starches [51]. In piglets, the transition from a milk diet to high carbohydrate nursery feed at weaning may explain the increase in *K. slooffiae* in post-weaned piglets. In comparison, *H. burtonii* is a common fungal contaminant of corn [52] and has not been found to be abundant in other piglet fecal studies regardless of piglet age [30,44]. However, *H. burtonii*, like *Candida*, is also able to degrade starch [53] and may compete with *K. slooffiae* in the gut environment. No differences in piglet growth rate or in bacterial composition were seen among the litters with high abundances of *H. burtonii*, indicating that high abundances of *H. burtonii* did not adversely affect piglet health. Further compositional, transient differences were seen between litters at early time points. One example is the temporary increase in *Dipodascaceae* in L.3 at day 3 (Figure 4B), which was present in high abundances in all pigs from that litter (data not shown). We hypothesize, that this temporary spike in *Dipodascaceae* unique to one litter reflects the highly transient nature of the early mycobiome as microbes compete for space and nutrients in the gastrointestinal tract as well as a potentially acute environmental exposure event by L.3 to *Dipodascaceae*, even though all litters were housed in the same environment.

Development had a strong effect on the associations among bacterial and fungal genera. Day 1 piglet fecal communities showed no interactions between fungus and bacterial genera, but as the fecal communities developed, there was a progressive increase in number and complexity of interactions between bacteria and fungus (Figure 7). This increase in interactions may be partially driven by a greater bacterial production of short-chain fatty acids following weaning. Firmicutes bacteria are known to produce a variety of SCFA via fermentation of complex carbohydrates that can inhibit fungal growth [54,55]. In particular, a negative correlation between SCFAs and potential fungal pathogen *Aspergillus* has been demonstrated in human diets [50] and several SCFAs have been shown to negatively correlate with relative abundances of *Aspergillus* in piglet digesta samples [5]. A similar negative association was seen in day 35 piglet feces in this study between *Aspergillus* and SCFA producers *Subdoligranulum*, *Prevotellaceae* UCG-003, and *Lachnospiraceae* UCG-010. Furthermore, previously published data on the mucosal microbiota of these piglets’ lower GI organs (colon and cecum) also showed significant negative correlations between *Aspergillus* and several SCFA producers, suggesting that SCFAs are likely inhibiting *Aspergillus* growth in the piglet gut [13]. While inferred interactions are seen, a lack of supporting literature on these fungal species makes further interpretation speculative and more studies are needed, but the effect of SCFA producers in the gut on the health of the piglet cannot be overlooked.

From birth through weaning, healthy piglet feces in this study showed a dramatic shift in the bacterial and fungal communities, as well as an increase in network connectivity between the two kingdoms. While the piglet fecal bacteriome followed a relatively stable and predictable pattern of development, the development of the mycobiome was more variable and likely more transient and environmentally driven, particularly prior to day 35 piglets. Due to the flexibility of mycobiome in early piglets and evidence from this study on the potential influence of fungal taxa found in feed, successful manipulation of the mycobiome with fungal probiotics may be beneficial during the weaning transition. Furthermore, potential fungal pathogens like *Aspergillus* may be suppressed in piglets by enhanced SCFAs. These SCFAs could be introduced through supplementation of SCFAs in the piglet diet or through the augmentation of bacterial populations that enhance SCFA production. Our findings provide a significant contribution to the understanding of the developing fecal mycobiome and potential bacterial-fungal interactions in healthy piglets. The ability of microbes to alter the environment and nutrient availability of other microbes is accepted, but these environmental changes can also alter the host health and growth. While studies in piglets are lacking, the direct role of microbes and their interactions and their ability to enhance piglet growth need to be assessed. Incorporating the mycobiome in future piglet microbiome studies will further elucidate the importance and role of fungi in porcine health.

## Figures and Tables

**Figure 1 microorganisms-08-00868-f001:**
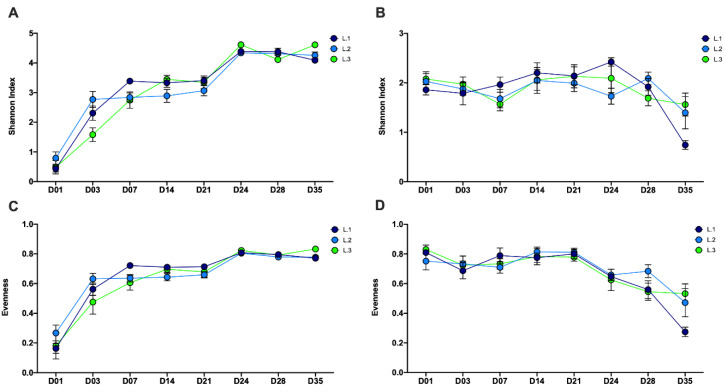
Alpha-diversity of the bacteriome and mycobiome in piglet feces over time. Shannon diversity index values for the (**A**) bacteriome and (**B**) mycobiome and evenness values for the (**C**) bacteriome and (**D**) mycobiome in piglet feces by sample time points (D01–35) and by litter (L.1–3).

**Figure 2 microorganisms-08-00868-f002:**
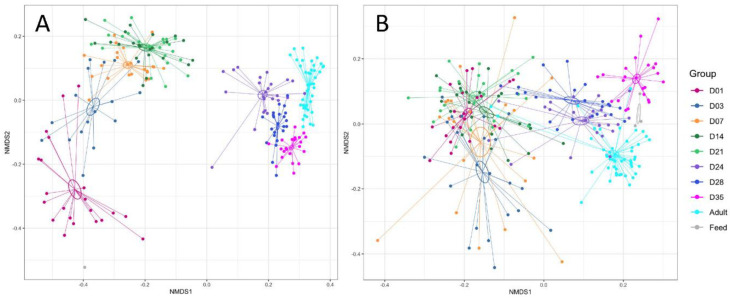
Beta diversity of piglet feces over time. Non-metric multidimensional scaling (NMDS) plot of β-diversity based on Bray–Curtis dissimilarities in the (**A**) bacteriome and (**B**) mycobiome of piglet feces for growth time points D01–35. Outdoor adult fecal samples (Adult), indoor lactating sow fecal samples (Sow) and nursery feed (Feed) are included for comparison only. Ellipses indicate 1 standard error from centroid. The effect of growth time points on piglet fecal communities was determined using permutational analysis of variance tests with the adonis function and strata option (strata = piglet) in the R package vegan.

**Figure 3 microorganisms-08-00868-f003:**
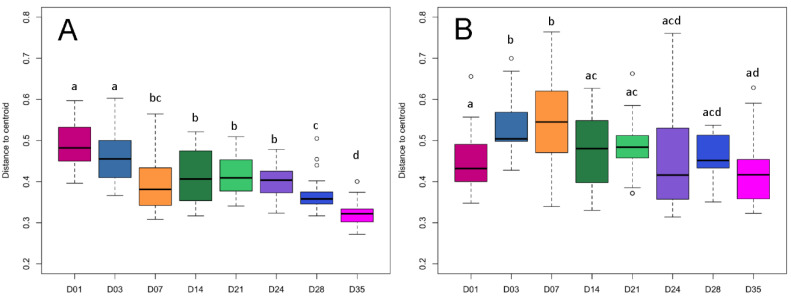
Boxplot of pairwise distances between piglet feces centroids over time. Plots represent the median and interquartile range in the (**A**) bacteriome and (**B**) mycobiome over time (D01–35). Differences between organ centroids were analyzed using permutational analysis of multivariate dispersion using the betadisper function in the R-package vegan on Bray–Curtis dissimilarities. Significance is indicated by letters a–d and open circles represent outliers (*p* < 0.05).

**Figure 4 microorganisms-08-00868-f004:**
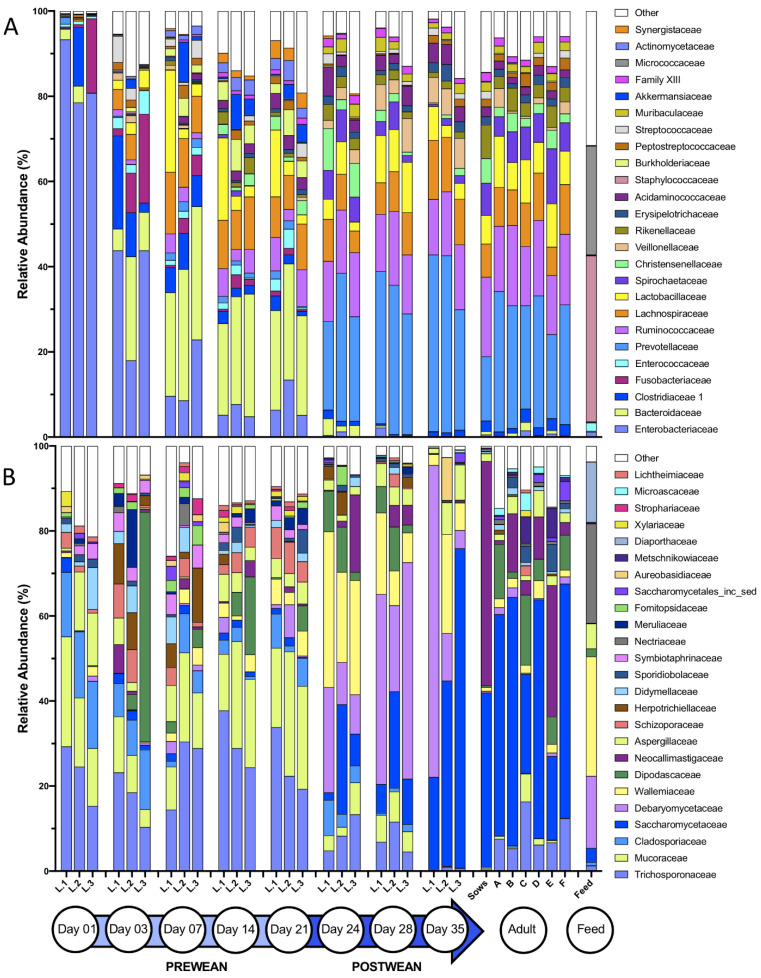
Taxonomic composition of the bacteriome and mycobiome in piglet feces over time. Mean percent relative abundances by litter (L.1–3) at the family level for the most abundant members of the (**A**) bacteriome and (**B**) mycobiome over time (D01–35). Sows (Sows), adult feces (A–F), and nursery feed (Feed) are included for comparison.

**Figure 5 microorganisms-08-00868-f005:**
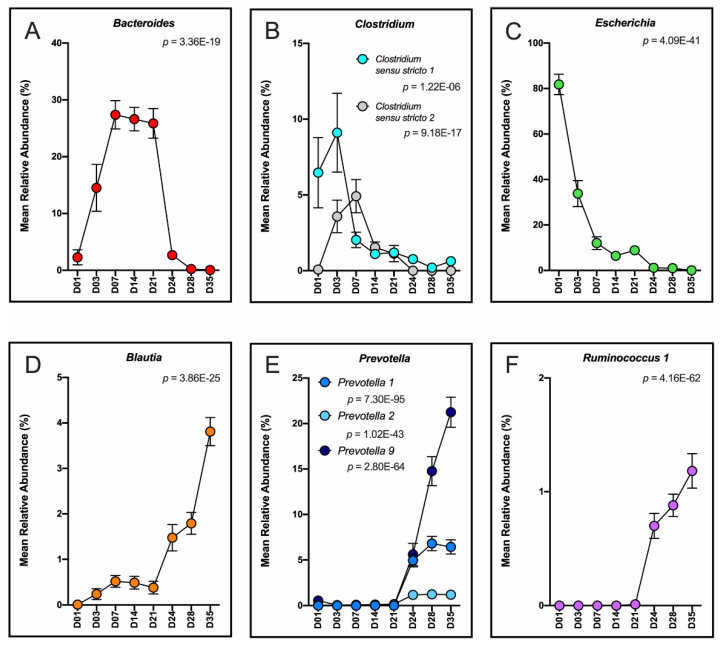
Differentially abundant genera in the piglet pre- vs. post-wean fecal bacteriome. Trends in mean relative abundances of significant bacterial genera representing the (**A**–**C**) pre-wean (D01–21) and (**D**–**F**) post-wean (D24–35) piglet fecal bacteriome. FDR-corrected p-values were calculated using a linear mixed-effect regression model in the R-package DAtest controlling for individual piglet and litter. Error bars represent ± SE. A complete list of differentially abundant bacteria is found in Appendix A.

**Figure 6 microorganisms-08-00868-f006:**
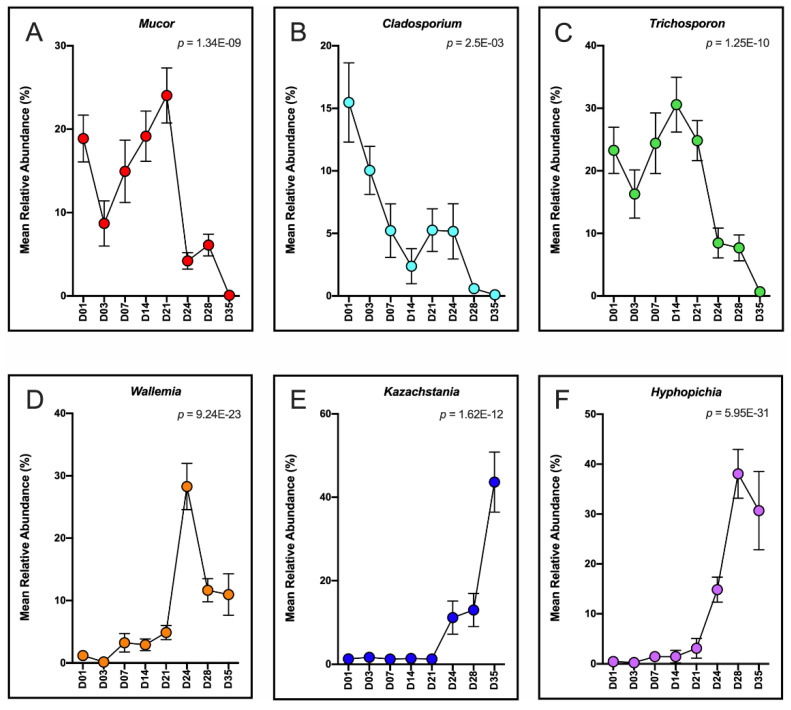
Differentially abundant genera over time in the piglet pre- vs. post-wean fecal mycobiome. Trends in mean relative abundances of significant fungal genera representing the (**A**–**C**) pre-wean (D01–21) and (**D**–**F**) post-wean (D24–35) piglet fecal mycobiome. FDR-corrected *p*-values were calculated using a linear mixed-effect regression model controlling for individual piglet. Error bars represent ± SE. A complete list of differentially abundant fungus is found in Appendix A.

**Figure 7 microorganisms-08-00868-f007:**
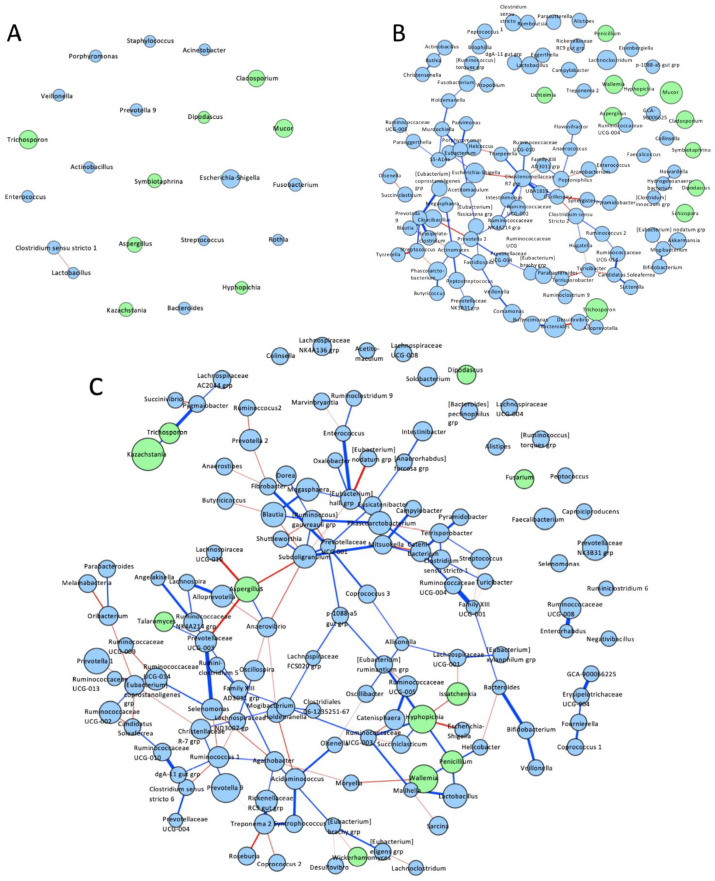
Network plots of bacterial and fungal associations in piglet feces over time. Network association plots between bacterial and fungal genera in piglet feces at (**A**) D01, (**B**) D21 and (**C**) D35. Associations were determined based on co-occurrence network analysis using the MB neighborhood selection method with the SPIEC-EASI R-package. The edge color indicates sign of correlation: negative (red), positive (dark blue); node color indicates kingdom: bacteria (light blue) and fungus (green). The size of node is proportional to the mean centered-log ratio abundance for each genus.

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
