# Peer review of "Temporal Dynamics of the Gut Bacteriome and Mycobiome in the Weanling Pig"

_microorganisms, 2020, doi:10.3390/microorganisms8060868_

Round 1

Reviewer 1 Report

The manuscript is well written and present interesting and novel data on the mycobiome succession profile in pigs in concert with the bacteriome profile (already well characterized). The analytical approach and interpretation focuses primarily on the mycobiome and associations of the mycobiome with the bacteriome. A few comments are provided to consider for improvement

L59 More information on the housing of piglets, particularly post weaning is required.  Provide information on the distribution among pens of pigs within litter, Was a single farrowing and nursery room used? Was the work conducted at a research facility of commercial farm (Would be good to have some idea of the nature of the facilities in terms of extent of exposure of piglets to possible environmental inoculants; particularly yeast inoculants).

L71. The comparison with outdoor adult sows is not justified and should be even if simply accessibility. Provide more detail on the nature and sampling of comparater pigs. Indicate number of outdoor sows sampled in each pen; total number of samples. Please indicate outdoor pigs were housed in pen "labeled A-F". Were the 2 lactating sows indoor sows?

L116  Correct: "....ASVs with <1 sequence in 1.0%.... 

L170 Figure 2. Why only 1 bacterial profile for feed? (3 samples taken)? Is it possible to enhance location of feed samples in Figure 2?  Profiles are note for "Adult" pigs. Does adult refer to outdoor adult pigs and the 2 lactating sows. Please indicate here and in M&M.

L189 Taxa vs. Taxon

L280 On  discussion linking low mycobiome diversity and health – fine to cite regarding relevance of mycobiome but I don’t see how this work contributes to the diversity argument (no comparators) as postulated in L284-285

L298 Much of the explanation regarding variation in the mycobiome is linked to their transient nature (allochthonous). Would there be value in querying/reporting the consistency with which a fungal ASV is recovered from the same pig over time. Does a fungal species colonize a single pig consistently or does it move about. I don’t think that can be inferred from the dispersion analysis but could add to discussion.

L315-317 Would be useful to confirm whether the two yeasts highlighted as showing striking variation in abundance were present in feed or not

L334 In the human study cited, humans were given fibre to enhance SCFA production. SCFA were not consumed.

L353 "...by the enhancing bacterial production..." I would be more comfortable if this statement first acknowledged the association found between SCFA and Aspergillus avoiding cause and effect implied here.

Author Response

Please see the attaachment.

Reviewer 2 Report

Arfken et al present a longitudinal characterization of healthy pigs microbiome (bacteriome and mycobiome) from birth to post-weaning based on 16S and ITS profiling from feces. The introduction clearly presents the domain and the rationale for such study. In particular, the longitudinal characterization of the pigs microbiome complements previous knowledge published from single time point studies. The material and method sections presents clearly the protocols, bacteria and yeast profiling and analysis of the data, using the recent, benchmarked and fully approved analysis pipeline DADA2. The results are clearly presented and highlight the increased complexity of the bacteriome, while the mycobiome is shows a strong decrease in complexity post-weaning. Finally, the authors present hypothesis to account for the evolution of the microbiome based on potential symbiosis or competition between specis, as well as a hypothesis on complementary impact of the respective evolutions of the bacteriome and mycobiome based on the known metabolic pathways of the microbiome. Globally the study constitutes the first, complete and well-documented description of the microbiome of the healthy pigs between birth and post-weaning, which constitutes a reference dataset for future studies aiming at investigating the impact of altered microbiome of pig health.

The following points should however be addressed before publication.

  • In the discussion, the interpretation of the metabolic impact of the evolution of the global microbiome is relatively shallow and focuses on a few diet related functions of the microbiome. Given the richness of the dataset and potential symbiotic versus completion between the bacterial and mycobiome populations, the authors should extend on the global impact on pig development (eg immunity, growth).
  • Figure 4 presents the taxonomic composition of the individual animals at all-time points. Despite a highly controlled protocol for feeding and nursing, some samples show interesting particularities (eg the presence of Fusobacteriacae in L3 day 1 for the bacteriome, or the presence of a large proportion of Dipodascaceae in L3 day 3 of the mycobiome. The authors should clarify these points and provide hypothesis to explain them.
  • While the profiling by 16S and ITS provides a mean to characterize the composition of the microbiome, the precise gene catalog required to further characterize the full metabolic potential of the microbiome and infer metabolic impacts of its evolution would require a more precise investigation by shotgun metagenomics, or at least a more thorough inference using for example PICRUST approach.
